# Occurrence of Candidemia in Patients with COVID-19 Admitted to Five ICUs in France

**DOI:** 10.3390/jof8070678

**Published:** 2022-06-28

**Authors:** Marion Blaize, Audrey Raoelina, Dimitri Kornblum, Laure Kamus, Alexandre Lampros, Marie Berger, Sophie Demeret, Jean-Michel Constantin, Antoine Monsel, Julien Mayaux, Charles-Edouard Luyt, Renaud Piarroux, Arnaud Fekkar

**Affiliations:** 1Sorbonne Université, INSERM, CNRS, Centre d’Immunologie et des Maladies Infectieuses, Cimi-Paris, Assistance Publique-Hôpitaux de Paris (AP-HP), Groupe Hospitalier La Pitié-Salpêtrière, Parasitologie Mycologie, F-75013 Paris, France; marion.blaize@aphp.fr; 2Assistance Publique-Hôpitaux de Paris (AP-HP), Groupe Hospitalier La Pitié-Salpêtrière, Parasitologie Mycologie, F-75013 Paris, France; audrey.raoelina@hotmail.fr (A.R.); dimitri.kornblum@aphp.fr (D.K.); laurekamus@aol.com (L.K.); alexandre.lampros@aphp.fr (A.L.); marie.berger@etu.sorbonne-universite.fr (M.B.); renaud.piarroux@aphp.fr (R.P.); 3Assistance Publique-Hôpitaux de Paris (AP-HP), Groupe Hospitalier La Pitié-Salpêtrière, Réanimation Neurologique, F-75013 Paris, France; sophie.demeret@aphp.fr; 4Sorbonne Université, GRC 29, Assistance Publique-Hôpitaux de Paris (AP-HP), Groupe Hospitalier La Pitié-Salpêtrière, Département d’Anesthésie Réanimation, F-75013 Paris, France; jean-michel.constantin@aphp.fr; 5Sorbonne Université, INSERM, Immunology Immunopathology Immunotherapy (I3), Assistance Publique-Hôpitaux de Paris (AP-HP), Groupe Hospitalier La Pitié-Salpêtrière, Département d’Anesthésie Réanimation, F-75013 Paris, France; antoine.monsel@aphp.fr; 6Assistance Publique-Hôpitaux de Paris (AP-HP), Groupe Hospitalier La Pitié-Salpêtrière, Réanimation Médicale, F-75013 Paris, France; julien.mayaux@aphp.fr; 7Sorbonne Université, INSERM, Institute of Cardiometabolism and Nutrition, Assistance Publique–Hôpitaux de Paris (APHP), Médecine Intensive Réanimation-Institut de Cardiologie, F-75013 Paris, France; charles-edouard.luyt@aphp.fr

**Keywords:** fungal infection, *Candida*, candidemia, candidiasis, fungemia, COVID-19, SARS-CoV-2

## Abstract

Whether severe COVID-19 is by itself a significant risk factor for the development of candidemia currently remains an open question as conflicting results have been published. We aim to assess the occurrence of candidemia in patients with severe COVID-19 admitted to the intensive care unit (ICU). We conducted a retrospective study on patients with severe SARS-CoV-2-related pneumonia admitted to 5 ICUs in France who were specifically screened for fungal complications between March 2020 and January 2021. The study population included a total of 264 patients; the median age was 56 years old and most of them were male (n = 186; 70.5%) and immunocompetent (n = 225; 87.5%), and 62.7% (n = 153/244) were on extracorporeal membrane oxygenation support. Microbiological analysis included 4864 blood culture samples and beta-glucan test performed on 975 sera. Candidemia was diagnosed in 13 (4.9%) patients. The species involved were mainly *C. albicans* (n = 6) and *C. parapsilosis* (n = 5). Almost all patients (12/13; 92.3%) had a colonization by yeasts. ICU mortality was not significantly impacted by the occurrence of candidemia. Unrelated positive beta-glucan tests were observed in 49 patients (23.4%), including 6 with mold infections and 43 with false positive results. In our series, patients with severe SARS-CoV-2-related pneumonia seemed at low risk of developing invasive candidiasis.

## 1. Introduction

Whether severe COVID-19 represents a significant risk factor for the development of invasive candidiasis remains an open question, as many reports indicating conflicting results have been published. In previous studies, the incidence of candidemia in severe COVID-19 patients varies between 0.8 and 14% [1,2]. In comparison with non-COVID-19 patients, a two- to ten-fold increase in candidemia has been reported in patients with severe COVID-19 [3,4,5].

During the first waves of the pandemic, patients who developed severe forms of SARS-CoV-2 infection were mainly immunocompetent with underlying chronic conditions such as hypertension, obesity, dyslipidemia, diabetes, or chronic obstructive pulmonary disease (COPD). None of these conditions constitute by themselves an obvious risk factor for developing invasive yeast infections. However, the link between severe COVID-19 and the occurrence of invasive candidiasis could be the result of one or more unrelated factors.

Many aspects of intensive care (e.g., exposure to broad-spectrum antibiotics or corticosteroids, invasive procedures, and implantation of devices such as central venous lines, intubation tubes, extra-corporeal-membrane oxygenation (ECMO)) and prolonged length of hospital stay have long been known as risk factors for developing fungal superinfections, especially invasive candidiasis [6,7,8,9,10].

The overwhelming surge of patients admitted to intensive care during this period, combined with staff shortages and exhaustion, may have also impacted the strict application of hygiene measures and may have led to an increased risk of nosocomial infection, particularly fungal infection [11]. It has also been suggested that SARS-CoV-2, which infects a large number of cell types and organs, can by itself lead to a greater susceptibility to yeast superinfections through its effect on the digestive tract, which harbors the commensal *Candida* spp. flora [12,13].

Additionally, deficiencies in the interferon pathway response (either inherited or related to the presence of anti-interferon autoantibodies) represent a very important risk of developing severe forms of COVID-19 [14]. Interactions have been described between the interferon pathways and the Toll-like receptor pathways, and recent data suggest a consistent role of type I interferon in the anti-*Candida* immune response [15]. 

Taking all these data into account, we aimed to retrospectively assess the incidence rate, epidemiological characteristics, and outcome of ICU-acquired invasive candidiasis in patients with severe COVID-19 admitted to intensive care.

## 2. Materials and Methods

### 2.1. Design and Patients

We performed a retrospective analysis in five independent ICUs from a 1850-bed tertiary care center, the La Pitié-Salpêtrière hospital, located in Paris, France. From March 2020 to January 2021, all patients with a confirmed diagnosis of SARS-CoV-2 infection (i.e., with a positive RT-PCR) who were admitted to the ICU for at least 48 h because of severe COVID-19 were included in the study. The primary goal was to assess the incidence rate and epidemiological characteristics of ICU-acquired invasive candidiasis in this severe and closely screened cohort of patients.

### 2.2. Case Definition

Proven candidemia was defined by the presence of one or more *Candida* species in at least one blood culture. 

### 2.3. Mycological Analysis

Blood cultures were performed using the system BACT/ALERTFA Plus (bioMérieux, Marcy l’Etoile, France) and, for some patients, the specific BACTEC Mycosis IC/F system (Becton Dickinson, Franklin Lakes, NJ, USA). Any positive sample was subjected to direct examination and subcultured on chromogenic agar (CHROMagar *Candida*; Becton Dickinson) to check for potential mixed infections. Subcultures were identified by mass spectrometry (Bruker Microflex, Bruker, Billerica, MA, USA) using the MSI-2 (Mass Spectrometry Identification) online database (https://msi.happy-dev.fr/, accessed on 6 May 2022). Serum sample analysis consisted of dosing the β-D-glucan by using the Fungitell assay in duplicate (Associates of Cape Cod, East Falmouth, MA, USA). The result was considered positive if two consecutive tests came back above a cutoff of 80 pg/mL.

### 2.4. Statistical Analysis

Statistical analysis was performed using the BiostaTGV website and the R version 3.5.3 statistical software (R Foundation for Statistical Computing). Continuous and categorical variables are presented as medians with interquartile range (IQR) or an absolute number with percentage. Categorical variables were compared using the Chi-square or the Fisher’s exact test. The student’s t-test was used for continuous variables. Due to the exploratory nature of this study, no Bonferroni correction was performed, and a two-sided α of less than 0.05 was considered statistically significant.

## 3. Results

### 3.1. Patient Characteristics

Over the 11-month period, 264 patients infected by SARS-CoV2 were admitted to intensive care and specifically screened for fungal superinfections (Appendix A). The median age of the study population was 56 years (IQR, 48–64 years). Most patients presented a “metabolic” phenotype, a known risk factor for severe COVID-19: they were predominantly male (n = 186; 70.5%), suffering from chronic hypertension (n = 138; 53.7%), diabetes (n = 88; 34%), and dyslipidemia (n = 50; 19.4%). Among the 257 patients for whom the information was available, 32 (12.5%) were immunocompromised and presented risk factors for invasive fungal infections as defined by EORTC/MSG [16]. The main risk factor was the administration of corticosteroid therapy or other immunosuppressive drugs. Eighteen patients were solid organ transplant (SOT) recipients, eight patients had hematological malignancies, and eight received prolonged corticosteroid therapy for autoimmune diseases. The study population was characterized by very severe respiratory presentation; all patients were intubated and placed on mechanical ventilation with severe respiratory failure (worst PaO_2_/FiO_2_; median, 60; IQR, 51–80). The median of the Simplified Acute Physiology Score II (SAPS II) was 55 (IQR, 39–68), a value corresponding to >50% of predicted hospital mortality. Therefore, many patients (n = 153/244; 62.7%) were placed on venovenous ECMO (vv-ECMO) support, and most of them needed vasopressor support with more than 0.5 mg/h of noradrenaline (n = 172/233; 73.8%). The median length of stay in the ICU was 30 days (IQR, 19–51). The overall mortality was 43% (n = 105/244). 

### 3.2. Blood Cultures and Mycological Tests Results

One hundred and sixty-four patients were colonized with yeast (respiratory tract and/or cutaneous and/or digestive tract and/or urinary tract). The most frequently identified species was *C. albicans*, in 123 patients (75%), followed by *C. glabrata* and *C. parapsilosis,* in 14 patients each (8.5%).

For almost all patients, at least one blood culture was drawn (261/264; 98.9%) amounting to a total of 4883 vials (mean: 18.5 per patient; minimum = 0, maximum = 132). Among the patients tested, 175 (67%) had at least one positive blood culture, either with a fungal agent for 13 patients, or with bacteria for 173 patients. Eleven patients had an episode of fungemia following or preceding an episode of bacteremia. It should be noted that we did not observe any co-infection. Finally, the number of positive vials for yeast represented just 1% of the total number of collected vials (50/4883).

A total of 975 beta-glucan tests were performed for 209 patients. The test was positive for half of the patients who developed candidemia (n = 6/12) and for a quarter of the patients (49/197) who did not have candidemia, some of whom developed an invasive mold infection (n = 6 aspergillosis), and others who had no documented fungal complications (n = 43). Only one patient had concomitant diagnosis of invasive aspergillosis and candidemia (*C. krusei*); he presented a positive beta-glucan. 

### 3.3. Characteristics of Patients with Candidemia

A diagnosis of candidemia was established for 13 out of 261 patients (4.9%). The species involved in the cases included *C. albicans* (n = 6; 46.2%), *C. parapsilosis* (n = 5; 38.5%), *C. krusei* (n = 1; 7.7%), and *C. tropicalis* (n = 1; 7.7%) (Table 1). The median time from ICU admission to fungemia (determined by the day on which the first positive blood culture was sampled) was 24 days (IQR: 20–28 days).

Among patients with candidemia, only 50% of those tested around the time of diagnosis (i.e., within 5 days before/after the blood culture was sampled) had a positive serum beta-glucan test (Table 2). On the other hand, almost all patients (12/13; 92.3%) had a previous or concomitant colonization by yeasts of either the respiratory tract or the skin. In a univariate model analysis, colonization was confirmed as significantly associated with the occurrence of candidemia (*p* = 0.02). 

The length of stay in the ICU was also statistically associated with candidemia, with a median length of stay of 55 days versus 30 days for non-candidemic patients (*p* = 0.02). In this study population, no association was observed between candidemia and underlying conditions such as hypertension, dyslipidemia, or diabetes. A trend was found linking the occurrence of candidemia and the overall severity score (SAPS II) (*p* = 0.08). Finally, the requirement for vv-ECMO support was not found to be associated with an increased risk of candidemia (*p* = 1).

If we now consider these results in comparison with the epidemiology of candidemia occurring in the different intensive care units of our hospital just before the pandemic (July 2019–December 2019), the results are quite different. The number of cases was 22 over this 6-month period, which is higher than what we observed during the pandemic. The species distribution was as follows: *C. albicans* (11; 50%), followed by *C. glabrata* (n = 3; 13.6%), *C. parapsilosis*, *C. krusei*, and *C. dubliniensis* (n = 2 for each of these three species; 9.1%), and finally *C. tropicalis* (n = 1; 4.5%). However, the populations concerned are completely different, making it difficult to compare these results. The patients hospitalized in those ICUs before the COVID-19 pandemic were admitted for different pathologies and presented different comorbidities. Many of them had undergone surgery or had received solid organ transplants.

### 3.4. Treatment and Outcome of Patients with Candidemia

Following appropriate investigations (eye fundus, cardiac ultrasound), no chorioretinitis or endocarditis was diagnosed. It should be noted that a specific imaging search for dissemination in another organ, especially the liver and spleen, was not undertaken for any of the patients. None of the patients had been pre-exposed to an antifungal drug. Once the diagnosis was made, the patients received mainly an echinocandin-based treatment that was usually de-escalated to an azole. Three patients did not receive any antifungal treatment, two of whom died early after diagnosis. We did not find any significant difference in the overall ICU mortality between non-candidemic patients, 43.1% (100/232), and patients with candidemia, 41.7% (5/12; one patient was lost to follow up) (*p* = 1).

## 4. Discussion

Conflicting data exist regarding the incidence of candidemia among patients with severe COVID-19 [1,2,3,4,5]. In this study, we analyzed the occurrence of invasive candidiasis in a cohort of 264 patients during an 11-month period. Our results indicate a low incidence of candidemia.

This work has several limitations. It is retrospective and single-centered. Most analyses were univariate and were performed on a small number of cases. However, it should be noted that the number of samples dedicated to fungal search would not necessarily have been higher. It should also be noted that this study involves five independent units, some of which have specific features, such as exclusive orientation towards the use of ECMO support.

Previously published data on invasive candidiasis among severe COVID-19 patients indicated various results (an incidence of candidemia from 0.8 to 14%) with a trend suggesting increased candidemia incidence [1,2,17], especially in comparison to the historical non-COVID-19 cohort [5]. We report in this study an incidence of candidemia of 4.9% (n = 13, over 11-month pandemic period) for ICU patients with severe COVID-19. This number is lower than what we usually observed before the pandemic (n = 22, over 6-month period), which is probably mainly due to the differences in the patients’ underlying comorbidities, cause of admission, and turnover of patients in the ICU. The relatively low incidence of candidemia could be explained by the absence of additional risk factors for candidemia among patients with severe COVID-19, such as digestive surgery, parenteral nutrition, and immunodepression.

In our series, candidemia cases seemed evenly distributed over space and time and did not appear linked to a particular ICU nor to the massive influx of patients admitted during the first wave. The use of ECMO assistance was not found to be a risk factor for candidiasis in this study.

The diagnosis of invasive candidiasis is challenging due to the low sensitivity of blood cultures, the poor sensitivity and specificity of beta-glucan testing, and the low availability and performance of molecular biology approaches (PCR *Candida*), and the need to rely on deep sterile specimens or specific imaging studies (trans-thoracic ultrasound, ophthalmologic examination, and computed tomography scan).

In our series, beta-glucan testing was used extensively, but its benefit afterwards seems to us very questionable. Only half of the patients with candidemia tested positive at the time of fungemia, and only five patients tested positive prior to sampling. Of the patients who were screened for beta-glucan and did not develop candidemia, 25% (49/197) were tested positive. Of these patients, a few had developed an invasive pulmonary mold infection [18,19], but for the majority, we retained the result that it was a false positive.

Contrary to a previous study by Machado et al., only one isolate was found that was resistant to antifungal drugs [4]. Similarly, *Candida auris*, whose first described case occurred in our institution [20], was not found, although its incidence is high in some studies [21]. More strikingly, we did not observe any candidemia due to *C. glabrata*. This observation might be related to the fact that patients with severe COVID-19 did not have the risk factors usually associated with *C. glabrata* infections, especially digestive surgery or long-term pre-exposure to azoles drugs.

Although the data must be compared with great caution, the percentage of *C. parapsilosis* cases among the species responsible for candidemia was very different between the pre-COVID-19 period and the pandemic phase. As *C. parapsilosis* is a skin-commensal germ that can cause infections through an epidemic mode, its greater frequency during the pandemic could reflect a deficiency in care-related hygiene. In this sense, because of a transfer to another institution after the blood culture was taken, patient number 6 did not receive antifungal treatment for the *C. parapsilosis*-positive blood culture. She progressed favorably in the absence of treatment, which raises the question of contamination at the time of sampling, unless the improvement in her general condition allowed her to control the infection. In our institution, we have previously been confronted with outbreaks of azole-resistant *C. parapsilosis* [22], and it should also be noted that patient 5 was infected by an azole-resistant *C. parapsilosis* isolate belonging to the circulating clone that had struck our hospital. However, other patients were infected with susceptible isolates, which partly rules out the hypothesis of the epidemic spreading and inadequate hygiene.

## 5. Conclusions

In our series, patients with severe SARS-CoV-2-related pneumonia admitted to the ICU seemed at a low risk of developing candidemia. The factors that were found to be associated with the occurrence of candidemia were previous yeast colonization and length of ICU stay. Beta glucan testing lacked performance in the diagnosis of candidemia in this setting.

## Figures and Tables

**Table 1 jof-08-00678-t001:** Clinical and mycological characteristics of 13 ICU patients with severe COVID-19-related pneumonia.

	Patient N	1	2	3	4	5	6	7	8	9	10	11	12	13
Demographic characteristics	Sex/Age	F/62	M/55	M/57	M/40	M/67	F/59	M/62	F/17	M/53	M/78	M/59	M/58	M/38
	Pre-existing immune defect	None	Solid organ transplant	None	Hydro cortisone	None	None	None	Methyl prednisolone	Dexa methasone	Dexa methasone	None	None	Prednisone
	Underlying chronic diseases	HTN, Ob	HTN, Tab	None	HTN, Ob, Tab	None	HTN, Ob, DM	HTN, Ob, DM, Dlip	None	HTN, DM	HTN, COPD	HTN, DM	HTN, Ob	HTN
Clinical characteristics	Length of stay in the ICU, days	71	33	66	NA	37	59	29	50	85	28	6	81	77
	Amine support	Yes	Yes	Yes	Yes	Yes	Yes	Yes	Yes	Yes	Yes	Yes	Yes	No
	Dialysis	No	No	Yes	No	No	No	No	Yes	Yes	No	Yes	No	No
	Worst PaO_2_/FiO_2_	65	80	50	58	43	6	69	94	61	74	55	49	69
Laboratory findings	Previous yeast colonization (location)	*C. albicans* (resp.)	*C. albicans* (resp.)	*C. parapsilosis* (resp.)	*C. albicans* *C. krusei* (resp.)	No	*C. lusitaniae* (resp.)	*C. albicans* (resp.)	*C. albicans* (resp. + cut.)	*C. albicans* *C. glabrata* (resp.)	*C. parapsilosis* (resp.)	*C. tropicalis* (resp.)	*C. albicans* (resp.)	*C. glabrata* (resp.)
	Beta-glucan ^1^ pg/mL	>523	<60	<60	>523	<60	142	<60	>523	503	<60	93	73	<60
	Fungemia species	*C. albicans*	*C. albicans*	*C. parapsilosis*	*C. krusei*	*C. parapsilosis ^3^*	*C. parapsilosis*	*C. albicans*	*C. albicans*	*C. albicans*	*C. parapsilosis*	*C. tropicalis*	*C. albicans*	*C. parapsilosis*
	Blood culture ^2^	24 *Candida* 6 bacteria	1 *Candida* 4 bacteria	9 *Candida* 1 bacteria	1 *Candida* 1 bacteria	1 *Candida* 2 bacteria	1 *Candida* 0 bacteria	1 *Candida* 5 bacteria	7 *Candida* 14 bacteria	1 *Candida* 2 bacteria	1 *Candida* 6 bacteria	1 *Candida* 0 bacteria	1 *Candida* 3 bacteria	1 *Candida* 3 bacteria
Treatment and outcome	Time ICU to diagnosis, days	28	23	21	11	20	28	24	36	12	26	1	62	68
	Specific anti-fungal therapy	Cas then Flu	Cas then Flu	Cas then Flu	Cas then Vor	Cas	None	Cas	Cas then Flu	Cas then Flu	None	None	Cas	Cas then Flu
	Outcome, day 30 after the diagnosis	Alive	Alive	Alive	NA	Alive	Alive	Dead	Dead	Alive	Dead	Dead	Dead	Alive

^1^ serum sampled 5 days before/after the diagnosis; ^2^ total number of positive blood cultures vials and type of germ; ^3^ fluconazole resistant isolate: Minimum Inhibitory Concentration (MIC) > 256 mg/L (gradient concentration strip method, resistance threshold > 4 mg/L). Definition of abbreviations: NA = not available; HTN = hypertension; Ob = obesity; Tab = tabagism; DM = diabetes mellitus; Dlip = dyslipemia; COPD = chronic obstructive pulmonary disease; yeast colonization location:resp. = respiratory tract; cut. = cutaneous; Cas = caspofungin; Flu = fluconazole; Vor = Voriconazole.

**Table 2 jof-08-00678-t002:** Comparison between patients with candidemia and those without during severe COVID-19-related pneumonia.

		No Candidemia (n = 251)	Candidemia (n = 13)	Univariable OR (95% CI)	*p*-Value ^1^
Demographic characteristics and underlying condition	Age, median (IQR), year	56 (48–64)	58 (53–62)	-	0.98
	Male Gender, n (%)	176/251 (70)	10/13 (77)	1.4 [0.38; 5.31]	0.76
	Body Mass Index > 25 kg/m^2^, n (%)	174/238 (73)	6/13 (46)	0.32 [0.10; 0.99]	0.053
	Hypertension, n (%)	129/244 (53)	9/13 (69)	2.01 [0.60; 6.70]	0.39
	Diabetes, n (%)	82/246 (33)	6/13 (46)	1.71 [0.56; 5.25]	0.37
	Dyslipidemia, n (%)	49/245 (20)	1/13 (8)	0.33 [0.04; 2.60]	0.47
	Active smoker, n (%)	16/236 (7)	2/13 (15)	2.54 [0.51; 12.26]	0.24
Risk factors for invasive fungal infection	Preexisting host factor, n (%)^2^	30/244 (12)	2/13 (15)	1.3 [0.27; 6.15]	0.67
	Hemopathy, n (%)	8/244 (3)	0 (0)	-	1
	Hematopoietic stem cell, allograft, n (%)	2/244 (1)	0 (0)	-	1
	SOT, n (%)	17/244 (7)	1/13 (8)	1.11 [0.14; 9.05]	1
	Corticosteroid therapy > 0.3 mg/kg, n (%)	7/244 (3)	1/13 (8)	2.82 [0.32; 24.8]	0.34
Mycological tests	Yeast colonization	152/251 (61)	12/13 (92.3)	0.13 [0.02; 1.02]	**0.02**
	Beta-D-glucan > 80 pg/mL	49/197 (25)	6/12 (50)	3.02 [0.93; 9.80]	0.09
Inflammatory markers	C-reactive protein, median (IQR), mg/L	170 (70–275)	210 (139–331)	-	0.31
	Ferritine, median (IQR), mg/L	1766 (844–3271)	940 (675–2251)	-	0.32
	IL-6, median (IQR), pg/mL	144 (40–565)	551 (299–560)	-	0.23
ICU management and clinical characteristics	ICU stay, median (IQR), d	30 (17–50)	55 (32–73)	-	**0.02**
	Time from ICU admission to diagnosis, median (IQR), d	-	24 (20–28)	-	-
	SAPS II, median (IQR)	54 (39–67)	65 (53–79)	-	0.08
	Intubation period, median (IQR), d	26 (14–44)	38 (27–51)	-	0.14
	Worst P/F, median (IQR)	60 (51–80)	61 (55–69)	-	0.9
	Extracorporeal membrane oxygenation, n (%)	145/232 (63)	8/12 (67)	1.2 [0.35; 4.10]	1
	Vasopressor support, n (%)	160/220 (73)	12/13 (92)	4.5 [0.57; 35.36]	0.19
	Dialysis, n (%)	71/222 (32)	4/12 (33)	1.06 [0.31; 3.64]	1
Mortality in ICU, n (%)		100/232 (43.1)	5/12 (41.7)	0.94 [0.29; 3.05]	1

Definition of abbreviations: OR = odds ratio; CI = confidence interval; IQR = interquartile range; SOT = solid organ transplantation; ICU = intensive care unit; SAPS = Simplified Acute Physiology Score; P/F = PaO_2_/FiO_2._ Statistically significant value appears in bold. ^1^
*p*-value was calculated using either the Chi-square test or the Mann–Whitney test. ^2^ As defined jointly by the European Organization for Research and Treatment of Cancer and Mycosis Study Group (EORTC/MSG) according to Donnelly and colleagues [16].

## Data Availability

All data of the study are available by requesting the corresponding author.

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
