# Peer review of "Occurrence of Candidemia in Patients with COVID-19 Admitted to Five ICUs in France"

_jof, 2022, doi:10.3390/jof8070678_

Round 1

Reviewer 1 Report

This manuscript by Blaize et al, Manuscript ID; jof-1738957, entitled “Occurrence of Candidemia in Patients with COVID-19 admitted to the ICU” presents incidence rate, epidemiological characteristics, and outcome of ICU-acquired candidemia in patients with severe COVID-19 in France. The addressed subject is interesting and within the scope of the journal, but this manuscript contains some minor points.

First of all, the manuscript is very well-written, but some parts require English editing.

Herein are some comments to improve the manuscript:

Title

· It is recommended to insert “France” in the manuscript title.

Affiliations:

· Please superscript “1” in “Dimitri Kornblum1”.

Abstract

· Line 27: Please change “invasive candidiasis” to “Candidemia”, because you just survey candida blood infections.

· It is recommended to add the time period (month, year) of this study to the Abstract.

· It is recommended to write more about Candidemia cases in the results section than in the whole ICU population.

· Line 38: Please correct “SARS-CoV-2 related pneumonia” to “SARS-CoV-2-related pneumonia”.

1. Introduction

· Line 55. Please correct “broad spectrum” to “broad-spectrum”.

2. Materials and Methods

· Please correct “Bact/Alert” to “BacT/ALERT”.

· Please correct “Bactec” to “BACTEC”.

3. Results

· Line 110: Please delete “(March, 2020 to January, 2021)” in the manuscript.

· Line 150: Did you see candidemia and mold infection coinfection at the same time? Please clarify.

· Line 150: Please specify which mold infections were isolated in the manuscript.

· Line 148 and 175: Please correct “beta glucan” to “beta-glucan”.

· Line 215: Please correct “de-escaladed” to “de-escalated”.

· Line 216: Please correct “significative” to “significant ”.

Table 1. Characteristics of the study population

· It is recommended to transfer this table as a supplementary table 1, Because most of the information is given in Table 3.

· Please change “Gender, Male, n (%)” to “Male Gender, n (%)”.

· It is recommended to sort “Demographic characteristics and underlying conditions” based on Most frequency to lowest.

Table 2.

· It is recommended to combine sex and age to “Sex/Age” Because the table is very large.

· It is recommended to abbreviate “Underlying chronic diseases”.

· You mention “C. parapsilosis is a fluconazole-resistant strain in patient 5”. Please explain what method you used to identify the resistant species. Please clarify in the manuscript.

· It is recommended to add the location/ source of “Previous yeast colonization” after each Candida species for each patient to table 2.

· Please abbreviate antifungal drugs in the table.

Table 3.

· Please change “Sex Male, n (%)” to “Male Gender, n (%)”.

· It is recommended to sort “Demographic characteristics and underlying conditions” based on Most frequency to lowest.

· Please correct “p value” to “p-value”.

· Please correct “Chi2square test” to “Chi-square test”.

4. Discussion

· Line 243, 247, and 250: Please correct “beta glucan” to “beta-glucan”. 

Reviewer 2 Report

I congratulate the authors for the initiative of the study that seems to me of all the interest and I propose the following changes:

-I suggest improving the pathophysiological explanation of the possible relationship between infection by Sars cov 2 and Candidemia.

-Expand the explanation of the findings found (the finding of a low rate of candidemia contrasts with a population at such a high risk for this disease).

-I suggest proposing some hypotheses that explain the findings at the discretion of the authors and that could be contrasted in the future.

Author Response

I congratulate the authors for the initiative of the study that seems to me of all the interest

We thank you for your kind comments which particularly please us.

I propose the following changes:

-I suggest improving the pathophysiological explanation of the possible relationship between infection by Sars cov 2 and Candidemia.

The pathophysiological explanation of the possible relationship between SARS-CoV-2 infection and candidiasis was very hypothetical and in our study, it is actually not verified.

-Expand the explanation of the findings found (the finding of a low rate of candidemia contrasts with a population at such a high risk for this disease).

-I suggest proposing some hypotheses that explain the findings at the discretion of the authors and that could be contrasted in the future.

The population of concern has some risk factors for candidemia but a number of other factors are absent. We have added a sentence: « The relatively low incidence of candidemia could be explained by the absence among patients with severe COVID-19 of additional risk factors for candidemia, such as digestive surgery, parenteral nutrition, immunodepression ». (line 244)

Reviewer 3 Report

  The author aimed to investigate whether COVID-19 is a significant risk factor for the development of candidemia in the intensive care unit. I have the following suggestion:

1.     Undoubtedly COVID-19 is currently a hot topic in recent years. There have been several studies that found the high prevalence of invasive fungal infections in ICU patients with COVID-19 (reference no.1~5). I cannot find the conflicting results that deserve further studies or question this issue. The author should mention why we should assess whether candidemia is important in COVID-19 patients? Dose candidemia increase the risk of mortality in COVID-19 patients?

2.     The author mentioned that the risk factors of COVID-19 infections were immunocompetent patients with underlying chronic conditions (line 50). Does it mean that immunocompromised patients, such as cancer patients on chemotherapy, long-term use of steroid, or old aged patients are not at risk of severe COVID-19 infections? I do not this so.

3.     Although this is a multi-center study, the overall case number is not adequate to support the conclusion that patients with severe COVID-19 related pneumonia were at low risk of developing invasive candidiasis. If the final conclusion is COVID-19 patients had low risk of invasive candidiasis, what is the key message that the study can tell the reader?

4.     What is the cutoff point that the author considered the incidence 4.9% to be low risk? Of course 4.9% is not high, what if the result is 14.9%, will the author think it is high or low? I think the conclusion should be: in our series, patients with severe COVID-19 infection were at lower risk than previous studies, but………. (some information new that deserve readers pay attention to)

5.     If the incidence of candidemia was only 4.9% in the cohort, it makes no sense to compare COVID-19 patients with candidemia with those without candidemia (table 3), and it is not necessary to have list all the 13 patients in Table 2. To be straight, if there were only 13 patients that had candidemia, why should I notice them?

6.     What is the key message of this study? Just to tell the readers that COVID-19 patients had low incidence of candidemia? Is there anything new in this study?

7.     Discussion: line 221, if there is conflicting data exist regarding the incidence of candidemia among patients with severe COVID-19, there should be references.

Author Response

Academic Editor felt that the report given by reviewer 3# was unqualified. However, please find below point-to-point responses.

Suggestions for Authors

The author aimed to investigate whether COVID-19 is a significant risk factor for the development of candidemia in the intensive care unit. I have the following suggestion:

  1. Undoubtedly COVID-19 is currently a hot topic in recent years. There have been several studies that found the high prevalence of invasive fungal infections in ICU patients with COVID-19 (reference no.1~5). I cannot find the conflicting results that deserve further studies or question this issue. The author should mention why we should assess whether candidemia is important in COVID-19 patients? Does candidemia increase the risk of mortality in COVID-19 patients?

Baddley et al (reference 1) analyzed 24 articles reporting candidemia in COVID-19 patients: they reported incidence ranging from 0.4 to 14%. We consider this gap as conflicting results and we wanted to know what incidence in our center was. Candidemia are associated with high mortality. They can be managed with targeted antifungal treatments that improve prognosis. We cannot (from the numbers presented in our study) judge the potential excess mortality related to candidemia since we highlight a low incidence of candidemia in patients with severe COVID-19

  1. The author mentioned that the risk factors of COVID-19 infections were immunocompetent patients with underlying chronic conditions (line 50). Does it mean that immunocompromised patients, such as cancer patients on chemotherapy, long-term use of steroid, or old aged patients are not at risk of severe COVID-19 infections? I do not this so.

Known risk factors of  severe COVID-19 are as hypertension, obesity or diabetes. But, of course, other patients (such as immunocompromised patients) could be infected by SARS-CoV-2.

  1. Although this is a multi-center study, the overall case number is not adequate to support the conclusion that patients with severe COVID-19 related pneumonia were at low risk of developing invasive candidiasis. If the final conclusion is COVID-19 patients had low risk of invasive candidiasis, what is the key message that the study can tell the reader?

The message is severe COVID-19 is not a risk factor for occurrence of candidemia.

  1. What is the cutoff point that the author considered the incidence 4.9% to be low risk? Of course 4.9% is not high, what if the result is 14.9%, will the author think it is high or low? I think the conclusion should be: in our series, patients with severe COVID-19 infection were at lower risk than previous studies, but………. (some information new that deserve readers pay attention to)

We acknowledge that it is useful to put incidence values into perspective with respect to known data, rather than deciding ex abrupto that a value is "high" or "low"; any absolute denomination being imperfect and imprecise. This is why we compare our data with published data and consider that the incidence observed in our center is low.

  1. If the incidence of candidemia was only 4.9% in the cohort, it makes no sense to compare COVID-19 patients with candidemia with those without candidemia (table 3), and it is not necessary to have list all the 13 patients in Table 2. To be straight, if there were only 13 patients that had candidemia, why should I notice them?

It is interesting to detail cases in order to analyse clinical data and try to understand why this patient had candidemia.

  1. What is the key message of this study? Just to tell the readers that COVID-19 patients had low incidence of candidemia? Is there anything new in this study?

Yes, it is about reporting a low incidence of candidemia in patients with severe COVID-19. Many articles report a much higher incidence. It is therefore important to bring to the attention of the international community the existence of totally different data with potential consequences in the management of patients with severe COVID-19

  1. Discussion: line 221, if there is conflicting data exist regarding the incidence of candidemia among patients with severe COVID-19, there should be references.

These are references 1-5, reported in the introduction (we have added these references on line 221)

Reviewer 4 Report

1.         The manuscript under peer-review entitled, “Occurrence of candidemia in patients with COVID-19 admitted to the ICU” assesses retrospectively the incidence rate, epidemiological characteristics and outcome of ICU-acquired candidiasis among patients with severe COVID19.

2.         The manuscript drafting, as far as the language is concerned, is satisfactory. The subject matter in the manuscript has been introduced in a precise manner, which deals with the related clinical findings of invasive candidiasis.

3.         The authors did not find any significant difference in the overall ICU mortality between non-candidemic patients, 43.1% and patients with candidemia, 41.7% (p=1). 

4.         There is conflicting data exists regarding the incidence of candidemia among patients with severe COVID19. In this study, the authors have analyzed the occurrence of invasive candidiasis in a cohort of 264 patients during an 11-month Covid period with the results indicating a low incidence of candidemia.

5.         The authors report an incidence of candidemia of 4.9% for ICU patients with severe COVID19. This number is lower than what they usually observed before the pandemic, which is probably mainly due to the differences in the patient's underlying comorbidities, cause of admission and turnover of patient in ICU.

6.         In this study, candidemia cases seemed evenly distributed over space and time, did not appear linked to a particular ICU nor to the massive influx of patients admitted during the first wave. Moreover, the use of ECMO assistance was not found to be a risk factor for candidiasis.

7.         In this series, patients with severe SARS-CoV-2 related pneumonia admitted to the ICU seemed at low risk of developing candidemia. The factors found associated with the occurrence of candidemia were previous yeast colonization and ICU length of stay. Moreover, beta glucan testing laked performance in the diagnoses of candidemia in this type of setting due to its false positivity.

8.         The study design and modus operandi are narrated at length in the manuscript dealing with invasive candidiasis. The standard protocol of conventional and molecular methods, have been adopted for detection, identification and treatment of candidemia.

9.         The Results are nicely depicted in Tabulated form. All the isolates were belonging to Candida genus and other ones have not been included in this evaluation. Most of the relevant statistical analyses are applied to draw the conclusion.

10.      As it is a retrospective study, there are certain limitations inherently ingrained in the manuscript, which could have been minimized if it was a prospective study. The authors have also admitted the fact themselves that this study has many limitations being retrospective and single-centered. Most analysis were univariate and done on a small number of cases. The number of patients should have been more to have unbiased results. Moreover, this study involves 5 independent units, some of which have specific features, such as exclusive orientation towards the use of ECMO support.

11.      The is Discussion focused on the findings of this manuscript, in the light of contemporary studies, which have been cited in the text. Some of the cited studies are having somewhat similar results as that of the current study.

12.      The References are adequate in number and tried to be justified with citations vis-a-vis the clinical findings of all the patients mentioned in the manuscript as well as comparing the contemporary studies by other workers on the subject matter.

Author Response

Response to reviewer 4 comments

Comments and Suggestions for Authors

  1. The manuscript under peer-review entitled, “Occurrence of candidemia in patients with COVID-19 admitted to the ICU” assesses retrospectively the incidence rate, epidemiological characteristics and outcome of ICU-acquired candidiasis among patients with severe COVID19.
  2. The manuscript drafting, as far as the language is concerned, is satisfactory. The subject matter in the manuscript has been introduced in a precise manner, which deals with the related clinical findings of invasive candidiasis.
  3. The authors did not find any significant difference in the overall ICU mortality between non-candidemic patients, 43.1% and patients with candidemia, 41.7% (p=1).
  4. There is conflicting data exists regarding the incidence of candidemia among patients with severe COVID19. In this study, the authors have analyzed the occurrence of invasive candidiasis in a cohort of 264 patients during an 11-month Covid period with the results indicating a low incidence of candidemia.
  5. The authors report an incidence of candidemia of 4.9% for ICU patients with severe COVID19. This number is lower than what they usually observed before the pandemic, which is probably mainly due to the differences in the patient's underlying comorbidities, cause of admission and turnover of patient in ICU.
  6. In this study, candidemia cases seemed evenly distributed over space and time, did not appear linked to a particular ICU nor to the massive influx of patients admitted during the first wave. Moreover, the use of ECMO assistance was not found to be a risk factor for candidiasis.
  7. In this series, patients with severe SARS-CoV-2 related pneumonia admitted to the ICU seemed at low risk of developing candidemia. The factors found associated with the occurrence of candidemia were previous yeast colonization and ICU length of stay. Moreover, beta glucan testing laked performance in the diagnoses of candidemia in this type of setting due to its false positivity.
  8. The study design and modus operandi are narrated at length in the manuscript dealing with invasive candidiasis. The standard protocol of conventional and molecular methods, have been adopted for detection, identification and treatment of candidemia.
  9. The Results are nicely depicted in Tabulated form. All the isolates were belonging to Candida genus and other ones have not been included in this evaluation. Most of the relevant statistical analyses are applied to draw the conclusion.
  10. As it is a retrospective study, there are certain limitations inherently ingrained in the manuscript, which could have been minimized if it was a prospective study. The authors have also admitted the fact themselves that this study has many limitations being retrospective and single-centered. Most analysis were univariate and done on a small number of cases. The number of patients should have been more to have unbiased results. Moreover, this study involves 5 independent units, some of which have specific features, such as exclusive orientation towards the use of ECMO support.
  11. The Discussion is focused on the findings of this manuscript, in the light of contemporary studies, which have been cited in the text. Some of the cited studies are having somewhat similar results as that of the current study.
  12. The References are adequate in number and tried to be justified with citations vis-a-vis the clinical findings of all the patients mentioned in the manuscript as well as comparing the contemporary studies by other workers on the subject matter.

Dear Reviewer,

we are grateful to you for your particularly attentive reading of our work. Your very detailed summary of the paper does not require any particular response.